



# 1 Atmospheric Oxidation Mechanism and Kinetics of Indole Initiated
# 2 by ·OH and ·Cl: A Computational Study

Jingwen Xue[1#], Fangfang Ma[1]*, Jonas Elm[2], Jingwen Chen[1], Hong-Bin Xie[1]*
[1]Key Laboratory of Industrial Ecology and Environmental Engineering (Ministry of Education), School of Environmental
Science and Technology, Dalian University of Technology, Dalian 116024, China
[2]Department of Chemistry and iClimate, Aarhus University, Langelandsgade 140, DK-8000 Aarhus C, Denmark
*Correspondence to*: Fang-Fang Ma (maff@dlut.edu.cn); Hong-Bin Xie (hbxie@dlut.edu.cn)
**Abstract.** The atmospheric chemistry of organic nitrogen compounds (ONCs) is of great importance for understanding the
formation of carcinogenic nitrosamines and ONC oxidation products might influence atmospheric aerosol particle formation
and growth. Indole is a polyfunctional heterocyclic secondary amine with global emission quantity almost equivalent to that
of trimethylamine, the amine with the highest atmospheric emission. However, the atmospheric chemistry of indole remains
unclear. Herein, the reactions of indole with ·OH/·Cl, and subsequent reactions of resulting indole-radicals with $O_2$ under 200
ppt NO and 50 ppt $HO_2$· conditions, were investigated by a combination of quantum chemical calculations and kinetics
modeling. The results indicate that ·OH addition is dominant pathway for the reaction of ·OH with indole. However, both ·Cl
addition and H-abstraction are feasible for the corresponding reaction with ·Cl. All favorably formed indole-radicals further
react with $O_2$ to produce peroxy radicals, which mainly react with NO and $HO_2$· to form organonitrates, alkoxy radicals and
hydroperoxide products. Therefore, the oxidation mechanism of indole is distinct from that of previously reported amines,
which primarily form highly oxidized multifunctional compounds, imines or carcinogenic nitrosamines. In addition, the peroxy
radicals from the ·OH reaction can form N-(2-formylphenyl)formamide ($C_8H_7NO_2$), for the first time providing evidence for
the chemical identity of the $C_8H_7NO_2$ mass peak observed in the ·OH + indole experiments. More importantly, this study is
the first to demonstrate despite forming radicals by abstracting an H-atom at the N-site, carcinogenic nitrosamines were not
produced in the indole oxidation reaction.

## 23 1 Introduction

Volatile organic compounds (VOCs) play a central role in air quality and climate change as their transformations are
highly relevant for the formation of secondary organic aerosols (SOA), toxic air pollutants and ozone ($O_3$) (Ehn et al., 2014;
Karl et al., 2018; Lewis Alastair, 2018; Li et al., 2019; Khare and Gentner, 2018; Ji et al., 2018). Therefore, an accurate
description of the atmospheric transformation mechanism and kinetics of VOCs is essential to fully explore the global impacts
of VOCs. Despite massive effort to understand the atmospheric fate of VOCs, current mechanism-based atmospheric models
often underestimate SOA and $O_3$ formation quantity. Therefore, the emission inventories or reaction mechanism employed in





the models are either missing some vital primary VOCs or there remain unrevealed reaction mechanism of currently known
VOCs. Hence, it is crucial to identify unaccounted reaction pathways of known VOCs or transformation mechanism of
unconsidered VOCs with high concentrations.
Organic nitrogen compounds (ONCs) are a subgroup of VOCs that are widely observed in the atmosphere (Silva et al.,
2008). Until now, about 160 ONCs have been detected in the atmosphere, accounting for 10% of total gas phase nitrogen (Ge
et al., 2011; Silva et al., 2008). Due to the adverse effects of ONCs on air quality (formation of particles via acid-base reactions
or generation of toxic nitrosamines, nitramines, isocyanic acid and low volatile products via gas phase oxidation), the chemistry
of ONCs has gained significant attention in the recent years (Almeida et al., 2013; Chen et al., 2017; Lin et al., 2019; Nielsen
et al., 2012; Zhang et al., 2015; Xie et al., 2014; Xie et al., 2015; Xie et al., 2017; Ma et al., 2018a; Ma et al., 2021a; Ma et al.,
2019; Shen et al., 2019; Shen et al., 2020). Detailed transformation pathways of a series of ONCs including low-molecular-
weight alkyl amines (Nicovich et al., 2015; Xie et al., 2014; Xie et al., 2015; Ma et al., 2021a), aromatic aniline (Xie et al.,
2017; Shiels et al., 2021), heterocyclic amines (Sengupta et al., 2010; Ma et al., 2018a; Borduas et al., 2016; Ren and Da Silva,
2019) and amides (Xie et al., 2017; Borduas et al., 2016; Borduas et al., 2015; Bunkan et al., 2016; Bunkan et al., 2015) have
been investigated. These studies have shown that the functional groups connected to the $NH_x$ ($x = 0, 1, 2$) group highly affect
the reactivity of ONCs and eventually lead to their different atmospheric impacts. Therefore, the comprehensive understanding
the reaction mechanism of ONCs with various functional groups linked to the $NH_x$ group is of great significance to assess the
atmospheric impact of ONCs.
Indole is a polyfunctional heterocyclic secondary amine (Laskin et al., 2009). Atmospheric indole has various natural and
anthropogenic sources including vegetation, biomass burning, animal husbandry, coal mining, petroleum processing and
tobacco industry (Ma et al., 2021b; Cardoza et al., 2003; Yuan et al., 2017; Zito et al., 2015). The global emission of indole is
around 0.1 Tg yr$^{-1}$ (Misztal et al., 2015), which is almost equivalent to that of trimethylamine (~ 0.17 Tg yr$^{-1}$) (Schade and
Crutzen, 1995; Yu and Luo, 2014) which has the highest emission among the identified atmospheric amines. A field
measurement study found that the concentration of indole can reach 1-3 ppb in ambient air during a springtime flowering event
(Gentner et al., 2014). From a structural point of view, the -NH- group of indole is located at 9-center-10-electron delocalized
π bonds, possibly altering its reactivity compared to that of previously well-studied aliphatic amines and aniline. Therefore,
considering the large atmospheric emission of indole and its unique structure compared to previously studied amines, the
reaction mechanism of indole needs to be further evaluated to assess its atmospheric impacts. Furthermore, elucidating the
reaction mechanism of indole will add to the fundamental understanding of the transformation mechanism of ONCs.
Hydroxyl radicals (·OH) are considered to be the most important atmospheric oxidants governing the fate of most organic
compounds (Macleod et al., 2007). Previous experimental studies have investigated the reaction kinetics ($k_{OH}$) and identified
the products of the ·OH + indole reaction. Atkinson et al. found that the $k_{OH}$ value of the ·OH + indole reaction is $1.54 \times 10^{-10}$
cm$^3$ molecule$^{-1}$ s$^{-1}$ at 298 K, translating to a 20 min lifetime of indole (Atkinson et al., 1995). Montoya-Aguilera et al. found
that isatin and isatoic anhydride are the two dominate monomeric products for ·OH initiated reaction of indole. More
importantly, they found that the majority of indole oxidation products can contribute to SOA formation with an effective SOA





yield of 1.3 ± 0.3 under the indole concentration (200 ppb) employed in their chamber study (Montoya-Aguilera et al., 2017).
Although the chemical formulas of some of the indole oxidation products have been detected, detailed mechanistic information
such as the products branching ratio of the ·OH initiated reaction of indole remains unknown. Additionally, the lack of
commercially available standards of some products presents a significant obstacle to identify the exact chemical identity of the
products. Therefore, to fully understand the role of indole in SOA formation, it is essential to investigate the detailed
atmospheric transformation of indole initiated by ·OH.

Besides reactions with ·OH, reactions with chlorine radicals (·Cl) have been proposed to be an important removal pathway

for ONCs due to the identification of new ·Cl continental sources and the high reactivity of ·Cl (Wang et al., 2022; Li et al.,
2021; Jahn et al., 2021; Xia et al., 2020; Young et al., 2014; Faxon and Allen, 2013; Riedel et al., 2012; Atkinson et al., 1989;
Ji et al., 2013; Thornton et al., 2010; Le Breton et al., 2018). ·Cl initiated atmospheric oxidation of ONCs can lead to the
formation of N-centered radicals, once a strong 2-center-3-electron (2c-3e) bond complex has been formed between ·Cl and
$NH_x$ ($x$ = 1, 2). (Mckee et al., 1996; Xie et al., 2015; Xie et al., 2017; Ma et al., 2018a). The formed N-centered radicals can
further react with NO to form carcinogenic nitrosamines, increasing the atmospheric impact of ONC emissions (Xie et al.,
2014; Xie et al., 2015; Xie et al., 2017; Ma et al., 2018a; Ma et al., 2021a; Onel et al., 2014a; Onel et al., 2014b; Nielsen et al.,
2012; Da Silva, 2013). As a secondary amine, indole reactions with ·Cl has the possibility of forming N-centered radicals and
subsequently forming nitrosamines via the reaction with NO. Since the -NH- group of indole is embedded in a unique chemical
environment compared to previously well-studied ONCs, the reaction mechanism of ·Cl with indole remain elusive. In addition,
there are only a few studies concerning the reactions of polyfunctional heterocyclic ONCs with ·Cl.

In this work, we investigated the reaction mechanism and kinetics of indole initiated by ·OH and ·Cl by employing a

combination of quantum chemical calculations and kinetic modeling. The initial reactions of ·OH/·Cl + indole and the
subsequent reactions with $O_2$ of resulting intermediates were further investigated.

## 2 Computational Details

### 2.1 Ab Initio Electronic Structure Calculations

All the geometry optimizations and harmonic vibrational frequency calculations were performed at the M06-2X/6-

31+G(d,p) level of theory (Zhao and Truhlar, 2008). Intrinsic reaction coordinate calculations were performed to confirm the
connections of each transition state between the corresponding reactants and products. Single point energy calculations were
performed at the CBS-QB3 method based on the geometries at the M06-2X/6-31+G(d,p) level of theory (Montgomery et al.,
1999). The combination of the M06-2X functional and CBS-QB3 method has successfully been applied to predict radical-
molecule reactions (Guo et al., 2020; Ma et al., 2021b; Wang et al., 2018; Wang and Wang, 2016; Wu et al., 2015; Wang et al.,
2017; Fu et al., 2020). $T_1$ diagnostic (Table S2) values in the CCSD(T) calculations within the CBS-QB3 scheme for the
intermediates and transition states involved in the key reaction pathways were checked for multireference character. The $T_1$
diagnostic values for all checked important species in this work are lower than the threshold value of 0.045, indicating the





reliability of applied single reference methods (Rienstra-Kiracofe et al., 2000). In addition, similar to our previous studies, a
literature value of 0.8 kcal mol$^{-1}$ for the isolated ·Cl was used to account for the effect of spin-orbit coupling in the ·Cl + indole
reaction (Nicovich et al., 2015; Xie et al., 2017; Ma et al., 2018a). Atomic charges of indole and pre-reactive complexes in
the ·Cl + indole reaction are obtained by natural bond orbital (NBO) calculations (Reed et al., 1985). All calculations were
performed within the Gaussian 09 package (Frisch et al., 2009). Throughout the paper, the symbols "R, RC, PC, TS, IM and
P" stand for reactants, pre-reactive complexes, post-reactive complexes, transition states, intermediates and products involved
in the reactions, respectively, and their subscripts denote different species. In addition, "A//B" was used to present the
computational method, where "A" is the theoretical level for single point energy calculations and "B" is that for geometry
optimizations and harmonic frequency calculations.

### 105   2.2 Kinetics Calculations

The reaction rate constants for the reactions of ·OH/·Cl + indole and the subsequent reactions of resulting primary

intermediates were performed with the MultiWell-2014.1 and MESMER 5.0 program (Barker and Ortiz, 2001; Barker, 2001;
Glowacki et al., 2012), respectively. For the reactions with tight transition states, the Rice-Ramsperger-Kassel-Marcus (RRKM)
theory within the MultiWell-2014.1 or MESMER 5.0 program was used to calculate the reaction rate constants based on
energies and structures at the CBS-QB3//M06-2X/6-31+G(d,p) level of theory (Holbrook, 1996; Robinson, 1972). For
barrierless entrance pathways (from R to RC), the long-range transition-state theory (LRTST) with a dispersion force potential
within the MultiWell-2014.1 program (Barker and Ortiz, 2001) or Inverse Laplace Transformation (ILT) method within the
MESMER 5.0 program was employed to calculate the reaction rate constants (Rienstra-Kiracofe et al., 2000). Computational
details for performing LRTST and ILT calculation were described in our previous studies (Ma et al., 2021a; Ma et al., 2021b;
Guo et al., 2020; Ding et al., 2020b). The parameters used in the LRTST calculations and Lennard-Jones parameters of
intermediates estimated by the empirical method proposed by Gilbert and Smith (Gilbert, 1990) are listed in Table S3 and
Table S4, respectively. N$_2$ was selected as the buffer gas, and an average transfer energy of $\Delta E_d = 200$ cm$^{-1}$ was used to
simulate the collision energy transfer between active intermediates and N$_2$. For the reactions involving H-abstraction or H-
shift, tunneling effects were taken into account in all of the reaction rate constants calculations by using a one-dimensional
unsymmetrical Eckart barrier (Eckart, 1930), and were discussed in Supporting Information (SI).

### 121   3 Results and Discussion

### 122   3.1 Initial Reactions of Indole

In principle, ·OH and ·Cl could add to the unsaturated C=C bonds and phenyl group or directly abstract H-atoms

connected to either to a C-atom or the N-atom of indole. Considering the planar $C_s$ structure of indole, ·OH and ·Cl addition
to one side of indole was only considered here. However, although numerous attempts have been made, we failed to locate the
TSs and addition IMs of ·Cl addition to the C2, C3, C4, C7, C8 and C9 sites of indole, suggesting that such additions are in





fact unfeasible. Therefore, 7 H-abstraction pathways of ·OH and ·Cl, respectively, 8 ·OH-addition pathways and 2 ·Cl-addition
pathways were considered for the ·OH/·Cl + indole reactions. The schematic zero-point energy (ZPE) corrected potential
energy surfaces of ·OH/·Cl + indole reactions are presented in Figure 1.

As can be seen from Figure 1, each H-abstraction reaction pathway proceeds through a RC and PC, and the addition

pathways through a RC for the ·OH/·Cl + indole reactions. For the H-abstraction pathways, the activation energy ($E_a$) for the
-NH- group for both reactions are at least 2.0 kcal mol$^{-1}$ lower than the corresponding $E_a$ values for the -CH- groups. This
indicates that H-abstraction from the -NH- group forming $C_8H_6N$ radicals and $H_2O$/HCl is the most favorable among all the
H-abstraction pathways. In addition, the activation energy for the H-abstraction from the -NH- group in the ·Cl + indole
reaction is much lower than the corresponding ·OH + indole reaction. This is consistent with previously reported reactions of
other amines with ·OH and ·Cl (Ma et al., 2018a; Ma et al., 2021a; Xie et al., 2014; Xie et al., 2015; Xie et al., 2017; Tan et
al., 2021; Ren and Da Silva, 2019; Borduas et al., 2015).

For the addition reactions, the most favorable reaction site differs for the indole + ·OH and indole + ·Cl reactions. Among

all 8 ·OH addition pathways, ·OH addition to the C7 site of the C=C bond via $TS_{1-7}$ forming $IM_{1-7}$ is the most favorable
pathway. Different from the reaction with ·OH, the additions of ·Cl to the C5 and C6 sites to form $IM_{2-5}$ and $IM_{2-6}$, respectively,
are significantly more favorable. By comparing the $E_a$ values of the addition and H-abstraction pathways for both ·OH/·Cl +
indole reactions, it can be concluded that ·OH addition to the C7 site is the most favorable for the ·OH + indole reaction. All
the ·OH + indole hydrogen abstraction reactions have high energy barriers. However, the additions of ·Cl to the C5 and C6
sites as well as the -NH- H-abstraction are all favorable due to their very lower $E_a$ values for the ·Cl + indole reaction.

**Figure 1: Schematic ZPE-corrected potential energy surface for the reactions of indole + ·OH (A) and indole + ·Cl (B) at the CBS-QB3//M062X/6-31+g(d,p) level of theory. The total energy of the reactants indole + ·OH/·Cl are set to zero, respectively (reference state).**

Interestingly, we found that all the pathways for the indole + ·Cl reaction can proceed via a stable 2c-3e bonded RC, which is different from that of the ·OH + indole reaction. Among all 2c-3e bonded RCs, only $RC_{2-10}$ from the -NH- abstraction pathway is formed between the N-atom and ·Cl, while the others are formed between the C-atom and ·Cl. Note that $RC_{2-11}$,





which forms from C-atom and ·Cl, is the most stable among all the formed RCs in the ·Cl + indole reaction. To the best of our
knowledge, this is the first time that such a stable 2c-3e bonded RC has been identified between the C-atom and ·Cl. In addition,
the energy of $RC_{2-10}$ is higher than that of the traditional 2c-3e bonded RCs formed from alkylamine and ·Cl, which would
result from the delocalization of lone pair electrons of the N-atom. By analyzing the NBO charges of these nine RCs (Table
S3), we found that significant charge transfer occurs between ·Cl and indole. The charge at Cl atom for $RC_{2-5}$, $RC_{2-6}$, $RC_{2-10}$,
$RC_{2-11}$, $RC_{2-12}$, $RC_{2-13}$, $RC_{2-14}$, $RC_{2-15}$ and $RC_{2-16}$ are -0.35 $e$, -0.33 $e$, -0.31 $e$, -0.39 $e$, -0.35 $e$, -0.33 $e$, -0.39 $e$, -0.35 $e$ and -
0.33$e$, respectively, indicating that all RCs are charge-transfer complexes. Similar charge-transfer complexes were also found
in our previous study of the ·Cl + piperazine reaction (Ma et al., 2018a).
With the master equation theory, the overall rate constants ($k_{OH}$ and $k_{Cl}$) and branching ratios ($\Gamma$) of the ·OH/·Cl + indole
reactions were calculated at 298 K and 1 atm. The calculated $k_{OH}$ and $k_{Cl}$ values of indole are $7.9 \times 10^{-11}$ cm$^3$ molecule$^{-1}$ s$^{-1}$ and
$2.9 \times 10^{-10}$ cm$^3$ molecule$^{-1}$ s$^{-1}$, respectively. The calculated $k_{OH}$ value is close to the available experimental value of $1.5 \times 10^{-10}$
cm$^3$ molecule$^{-1}$ s$^{-1}$ (Atkinson et al., 1995), supporting the reliability of employed computational methods. Over the temperature
range 230-330 K (Ma et al., 2018b), the calculated $k_{OH}$ and $k_{Cl}$ values have a negative correlation with temperature (Figure S1).
Based on the calculated $\Gamma$ values of the ·OH/·Cl + indole reactions (Table 1), it can be concluded that $IM_{1-7}$ (77%) is the main
product for ·OH + indole reaction, and $IM_{2-5}$ (31%), $IM_{2-6}$ (46%) and $P_{2-10}$ ($C_8H_6N$ radicals + HCl) (23%) are the main products
for ·Cl + indole reaction. In addition, the calculated $\Gamma$ values of $IM_{1-7}$, $IM_{2-5}$, $IM_{2-6}$ and $P_{2-10}$ ($C_8H_6N$ radicals + HCl) change
negligibly with temperature in the range of 230-330 K (Figure S2). Therefore, we mainly considered the further transformation
of $IM_{1-7}$, $IM_{2-5}$, $IM_{2-6}$ and $C_8H_6N$ radicals in the following part.
**Table 1. Calculated branching ratios ($\Gamma$) for the indole + ·OH/·Cl reactions at 1 atm and 298 K.**

| Pathways | Species | $\Gamma$ | Species | $\Gamma$ | Species | $\Gamma$ |
|---|---|---|---|---|---|---|
| ·OH + Indole | $IM_{1-2}$ | 0 | $IM_{1-3}$ | 0 | $IM_{1-4}$ | 5% |
| | $IM_{1-5}$ | 12% | $IM_{1-6}$ | 3% | $IM_{1-7}$ | 77% |
| | $IM_{1-8}$ | 1% | $IM_{1-9}$ | 1% | $P_{1-10}$ | 1% |
| | $P_{1-11}$ | 0 | $P_{1-12}$ | 0 | $P_{1-13}$ | 0 |
| | $P_{1-14}$ | 0 | $P_{1-15}$ | 0 | $P_{1-16}$ | 0 |
| ·Cl + Indole | $IM_{2-5}$ | 31% | $IM_{2-6}$ | 46% | $P_{2-10}$ | 23% |
| | $P_{2-11}$ | 0 | $P_{2-12}$ | 0 | $P_{2-13}$ | 0 |
| | $P_{2-14}$ | 0 | $P_{2-15}$ | 0 | $P_{2-16}$ | 0 |

**3.2 Subsequent Reactions of Addition Intermediates**
Similar to other C-centered radicals (Zhang et al., 2012; Guo et al., 2020; Ma et al., 2021b; Yu et al., 2016; Yu et al., 2017;
Ji et al., 2017; Ding et al., 2020a), the intermediates $IM_{1-7}$, $IM_{2-5}$ and $IM_{2-6}$ will subsequently react with $O_2$. Two different
pathways (see Figure 2) were considered for the reactions of the intermediates $IM_{1-7}$, $IM_{2-5}$ and $IM_{2-6}$ with $O_2$. One is the direct





hydrogen abstraction by $O_2$ from the C site connecting to the -OH or -Cl group forming $P_{1-7-1}$ ($C_8H_7NO + HO_2\cdot$), $P_{2-5-1}$ ($C_8H_6NCl$
$+ HO_2\cdot$) and $P_{2-6-1}$ ($C_8H_6NCl + HO_2\cdot$). The other is the $O_2$ addition to the C sites with high spin density (see spin density
distribution in Table S4) of the intermediates $IM_{1-7}$, $IM_{2-5}$ and $IM_{2-6}$ to form peroxy radicals $Q$-$i$OO-$a/s$, where $Q$ stands for
intermediates $IM_{1-7}$, $IM_{2-5}$ and $IM_{2-6}$, $i$ stands for the numbering of the C-positions where $O_2$ is added. The $O_2$ molecule can be
added to the same (-$syn$, abbreviated as -$s$) and opposite (-$anti$, abbreviated as -$a$) sides of the plane relative to -OH or -Cl
group. The C2, C4, C6 and C8 sites of $IM_{1-7}$, C2, C4, C6 and C8 sites of $IM_{2-5}$ and C3, C5, C7 and C9 sites of $IM_{2-6}$ are high
spin density sites susceptible for $O_2$ addition.
As can be seen from the energetic data shown in Figure 2, $O_2$ addition to the C4 site of $IM_{1-7}$ to form $IM_{1-7}$-4OO-$a/s$ (-
0.6/-0.6 kcal mol$^{-1}$), C6 site of $IM_{2-5}$ to form $IM_{2-5}$-6OO-$a/s$ (-0.3/-2.0 kcal mol$^{-1}$) and C5 site of $IM_{2-6}$ to form $IM_{2-6}$-5OO-$a/s$
(2.0/1.7 kcal mol$^{-1}$) are the most favorable among all possible entrance pathways for the respective reactions. It deserves
mentioning that the formation energy ($\Delta E$) of $IM_{2-5}$-6OO-$a/s$ and $IM_{2-6}$-5OO-$a/s$ are only about 9.0 kcal mol$^{-1}$, which could
indicate that they likely re-dissociate back to the reactants $IM_{2-5}$/$IM_{2-6}$ and $O_2$, if $IM_{2-5}$-6OO-$a/s$ and $IM_{2-6}$-5OO-$a/s$ does not
rapidly transform to other species.
For the further transformation of the formed peroxy radicals $IM_{1-7}$-4OO(-$a/s$), $IM_{2-5}$-6OO(-$a/s$) and $IM_{2-6}$-5OO(-$a/s$), two
transformation pathways were identified. The first is cyclization reactions where the terminal O-atom of -OO attacks the
different C-positions to form bicycle radicals $Q$-$ij$OO(-$a/s$) ($j$ stands the number of the C-positions attacked by terminal O-
atom). The second is H-shifts from -OH, -NH- and different -CH- sites to the terminal O-atom to form various hydroperoxide
radicals $Q$-$i$OO-OH(-$a/s$), $Q$-$i$OO-NH(-$a/s$) and $Q$-$i$OO-C$k$H(-$a/s$) ($k$ stands the number of the C-positions from which H is
shifted), respectively. For $IM_{1-7}$-4OO(-$a/s$) and $IM_{2-5}$-6OO(-$a/s$), forming $IM_{1-7}$-4OO-OH-$s$ and $IM_{2-5}$-6OO-C5H-$a$ via H-shift
reactions are the most favorable, respectively. However, for $IM_{2-6}$-5OO(-$a/s$), the cyclization reaction forming $IM_{2-6}$-52OO-$a$
is the most favorable. It is noted that the formed $IM_{1-7}$-4OO-OH-$s$ from $IM_{1-7}$-4OO(-$a/s$) can barrierlessly transform to form
$C_8H_7NO_2$ (N-(2-formylphenyl)formamide) and $\cdot$OH (collectively denoted $P_{1-7-4-1}$) via concerted C-C and O-O bonds rupture.
The further transformation of the peroxy radicals $IM_{1-7}$-4OO(-$a/s$), $IM_{2-5}$-6OO(-$a/s$) and $IM_{2-6}$-5OO(-$a/s$) need to overcome
barriers above 20.5 kcal mol$^{-1}$ (relative to their respective peroxy radicals), indicating that the further transformation of $IM_{1-7}$-
4OO(-$a/s$), $IM_{2-5}$-6OO(-$a/s$) and $IM_{2-6}$-5OO(-$a/s$) should be very slow.











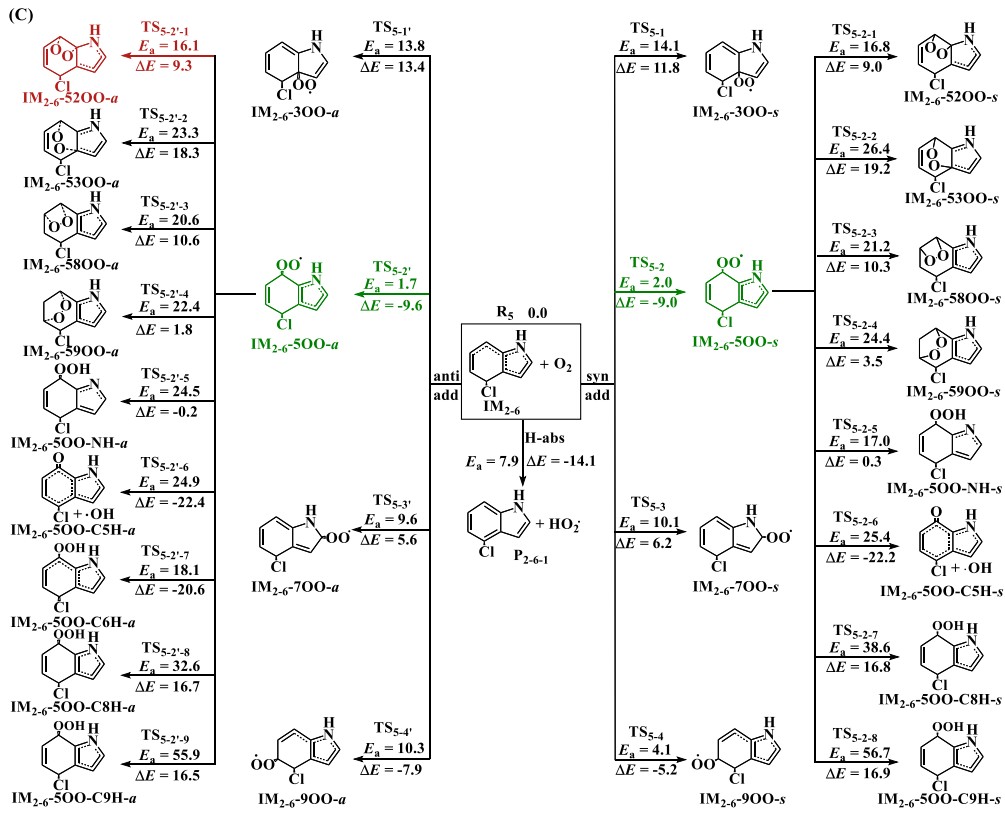


**Figure 2: Reaction pathways and corresponding energetic data for the reactions of $IM_{1-7}$ (A), $IM_{2-5}$ (B) and $IM_{2-6}$ (C) with $O_2$. Units are in kcal mol$^{-1}$.**

Based on the energetic data of the favorable reaction pathways, MESMER modeling was employed to investigate the reaction rate constants and fractional yields for the reactions of $IM_{1-7}$, $IM_{2-5}$, and $IM_{2-6}$ with $O_2$. Similar to previous studies (Guo et al., 2020; Ma et al., 2021a; Ma et al., 2021b; Zhang et al., 2012; Fu et al., 2020), bimolecular reactions with NO/HO$_2$· are considered as competitive pathways for the unimolecular reactions of the peroxy radicals $IM_{1-7}$-4OO(-$a/s$), $IM_{2-5}$-6OO(-$a/s$) and $IM_{2-6}$-5OO(-$a/s$) by simply adding their pseudo-first-order rate constants into the master equation modeling. Here, applied pseudo first order rate constants for peroxy radicals ($IM_{1-7}$-4OO(-$a/s$), $IM_{2-5}$-6OO(-$a/s$) and $IM_{2-6}$-5OO(-$a/s$)) reaction with NO and HO$_2$· are 0.06 s$^{-1}$ and 0.02 s$^{-1}$, respectively, corresponding to reactions occurring at 200 ppt NO and 50 ppt HO$_2$· conditions (Hofzumahaus et al., 2009; Yu et al., 2020; Praske et al., 2018). The reactions of peroxy radicals with NO and HO$_2$· should form organonitrate/alkoxy radicals (collectively denoted NO-P$_n$, where $n$ marks products from the different peroxy radical reactions) and hydroperoxide (HO$_2$-P$_n$), respectively. Pseudo-first-order rate constants of $IM_{1-7}$, $IM_{2-5}$, and $IM_{2-6}$ with $O_2$ are calculated to be $3.0 \times 10^7$ s$^{-1}$, based on the reaction rate constants of $IM_{1-7}$, $IM_{2-5}$, and $IM_{2-6}$ with $O_2$ ($6.0 \times 10^{-12}$ cm$^3$ molecule$^{-1}$ s$^{-1}$) and the concentration of $O_2$ ([$O_2$] = $5.0 \times 10^{18}$ molecule cm$^{-1}$). The simulated time-dependent fractional yields are presented in Figure 3.



218

219

**Figure 3: Calculated fractional yields of species (at 200 ppt NO and 50 ppt HO$_2$· conditions) as a function of time in the reactions of IM$_{1-7}$ (A), IM$_{2-5}$ (B), IM$_{2-6}$ (C) and C$_8$H$_6$N (D) with O$_2$ at 298 K and 760 Torr.**

As can be seen in Figure 3, after 100 s, the reactions of IM$_{1-7}$, IM$_{2-5}$ and IM$_{2-6}$ with O$_2$ mainly form the organonitrate/alkoxy radicals NO-P$_3$ (C$_8$H$_8$N$_2$O$_3$/C$_8$H$_8$NO$_2$·), NO-P$_4$ (C$_8$H$_7$N$_2$O$_3$Cl/C$_8$H$_7$NClO·) and NO-P$_5$ (C$_8$H$_7$N$_2$O$_3$Cl/C$_8$H$_7$NClO·), followed by the formation of hydroperoxide HO$_2$-P$_3$ (C$_8$H$_9$NO$_3$), HO$_2$-P$_4$ (C$_8$H$_8$NO$_2$Cl) and HO$_2$-P$_5$ (C$_8$H$_8$NO$_2$Cl), respectively. For the reactions of IM$_{2-5}$ and IM$_{2-6}$ with O$_2$, the main products formed are NO-P$_{4/5}$ and HO$_2$-P$_{4/5}$. In contrast, the IM$_{1-7}$ + O$_2$ reaction also lead to the fragmental products P$_{1-7-4-1}$ (C$_8$H$_7$NO$_2$ and ·OH) besides the main products NO-P$_3$ and HO$_2$-P$_3$. This difference in product branching ratios results from the lower unimolecular reaction energy barrier of the peroxy radicals IM$_{1-7}$-4OO(-*a/s*) from the reaction of IM$_{1-7}$ with O$_2$ than those of IM$_{2-5}$-6OO(-*a/s*) and IM$_{2-6}$-5OO(-*a/s*) from the reactions of IM$_{2-5}$ and IM$_{2-6}$ with O$_2$. It should be noted that the C$_8$H$_7$NO$_2$ product has been detected in previous experimental study of the ·OH + indole reaction (Montoya-Aguilera et al., 2017), supporting the validity of our computational results.





An obvious difference for these three reactions is that the reaction of $IM_{1-7}$ with $O_2$ can form peroxy radicals $IM_{1-7}$-4OO(-
*a/s*) with high yields during the reactions. However, the yields of the corresponding peroxy radicals $IM_{2-5}$-6OO(-*a/s*) and $IM_{2-}$
$_6$-5OO(-*a/s*) from the reactions of $IM_{2-5}$ and $IM_{2-6}$ with $O_2$ are low. The difference mainly originates from the difference in the
formation energy of these three peroxy radicals as shown in Figure 2. The $\Delta E$ values of $IM_{1-7}$-4OO(-*a/s*)(-19.1/-19.4 kcal mol$^{-}$
$^1$) are much more lower than those of $IM_{2-5}$-6OO(-*a/s*)(-9.0/-8.1 kcal mol$^{-1}$) and $IM_{2-6}$-5OO(-*a/s*)(-9.6/-9.0 kcal mol$^{-1}$). As
discussed above, the high formation energy of $IM_{2-5}$-6OO(-*a/s*) and $IM_{2-6}$-5OO(-*a/s*) should make $IM_{2-5}$-6OO(-*a/s*) and $IM_{2-6}$-
5OO(-*a/s*) return back to the reactants, explaining the reason for the lower yields of $IM_{2-5}$-6OO(-*a/s*) and $IM_{2-6}$-5OO(-*a/s*).
**3.3 Subsequent Reactions of $C_8H_6N$ radicals from the H-abstraction pathway**
Here, the biomolecular reaction with $O_2$ was mainly considered for $C_8H_6N$ radicals as its sole atmospheric fate. It was
found that the spin density distribution was mainly centered at the C atoms (C4 (0.662), C6 (0.261), C8 (0.178)) and N atom
(0.256), indicating that the $C_8H_6N$ radical is delocalized. This is in contrast to previously studied N-centered radicals formed
from alkylamines oxidation, which are highly localized (Xie et al., 2015; Xie et al., 2014; Ma et al., 2018a; Tan et al., 2021;
Borduas et al., 2015). Therefore, $O_2$ addition to the C4, C6, C8 and N1 sites (including attack from both sides) are considered
for the reaction of the $C_8H_6N$ radicals with $O_2$. As can be seen from Figure 4, $O_2$ additions to the C4 site of the $C_8H_6N$ radicals
forming $C_8H_6N$-4OO-*a/s* with $E_a$ of -0.3 kcal mol$^{-1}$ are the most favorable, translating to pseudo-first-order reaction rate
constants of $3.0 \times 10^7$ s$^{-1}$. Such rate constants are about 7 orders of magnitude higher than that of typical N-centered radicals
reacting with NO even under very high NO concentration (5 ppb). Therefore, $C_8H_6N$ radicals does not react with NO to form
carcinogenic nitrosamines in any appreciable amount, which is different from the previously reported reaction mechanism of
N-centered radicals formed from the reactions of alkylamines with ·Cl (Xie et al., 2015; Xie et al., 2014; Ma et al., 2018a). To
the best of our knowledge, this is the first study to reveal despite forming radicals by abstracting an H-atom at the N-site,
carcinogenic nitrosamines were not produced in the indole oxidation reaction.





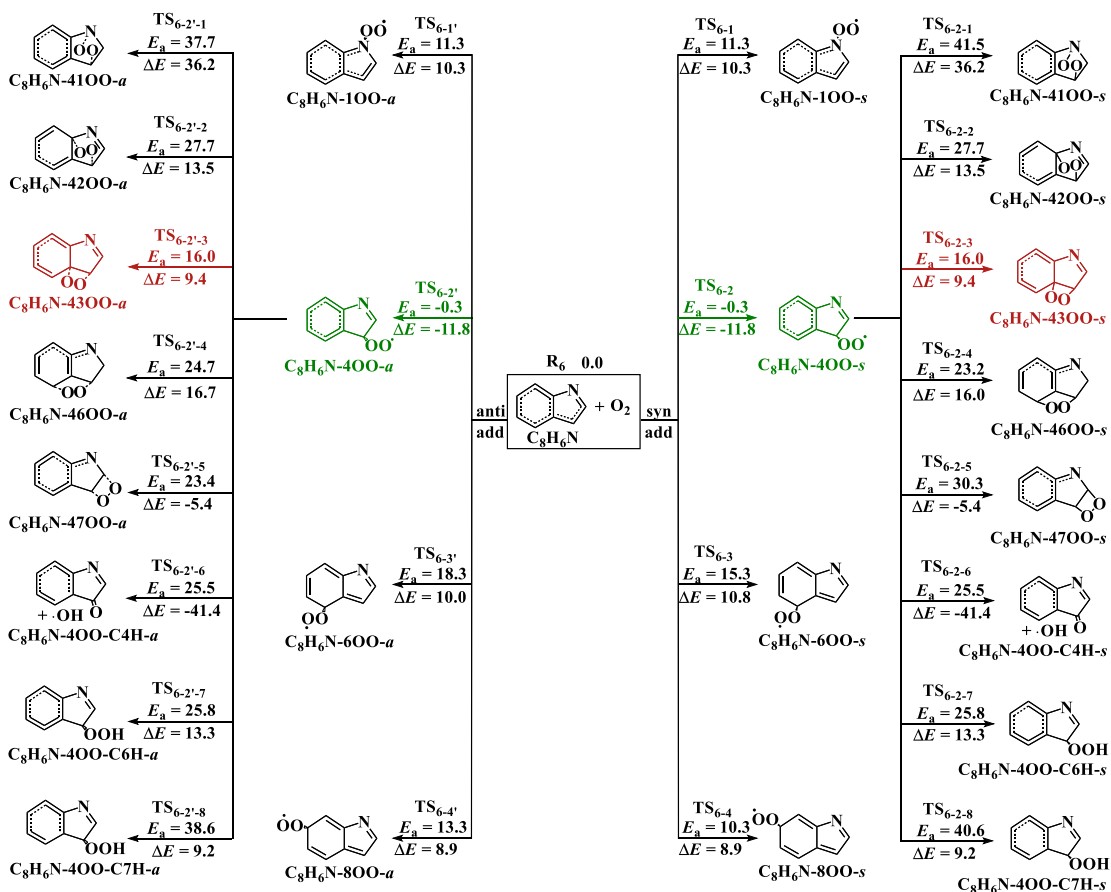


**Figure 4: Reaction pathways and corresponding energetic data for the reactions of $C_8H_6N$ radicals with $O_2$. Units are in kcal mol$^{-1}$.**

For the transformation of the formed $C_8H_6N$-4OO-*a/s* radicals, the ring closure reaction to form $C_8H_6N$-43OO-*a/s* is the most favorable, but still needs to overcome a 27.8 kcal mol$^{-1}$ energy barrier, therefore the further transformation of the formed $C_8H_6N$-4OO-*a/s* should proceed very slowly. The $C_8H_6N$-4OO-*a/s* should mainly react with NO and HO$_2$· to form NO-P$_6$ and HO$_2$-P$_6$. Detailed kinetics calculations (Figure 3D) further confirm that the reaction of $C_8H_6N$ radicals with $O_2$ mainly form NO-P$_6$ and HO$_2$-P$_6$ under 200 ppt NO and 50 ppt HO$_2$· conditions.

## 4 Comparison with Available Experimental Results and Atmospheric Implications.

This study found that ·OH/·Cl initiated reactions of indole mainly form organonitrates, alkoxy radicals and hydroperoxide products with N-(2-formylphenyl)formamide ($C_8H_7NO_2$) as a minor product at 200 ppt NO and 50 ppt HO$_2$· conditions. The formed closed-shell products have high oxygen-to-carbon ratios compared to indole and therefore are expected to have lower vapor pressures, likely being first generation products that can be further oxidized and contribute to the formation of SOA.



With our findings, a comparison was made with the available experimental study on ·OH initiated reaction of indole. The
calculated $k_{OH}$ values ($7.9 \times 10^{-11}$ cm$^3$ molecule$^{-1}$ s$^{-1}$) of indole is consistent with the experimental value ($15 \times 10^{-11}$ cm$^3$
molecule$^{-1}$ s$^{-1}$) (Atkinson et al., 1995), indicating the reliability of applied theoretical methods. A signal with the molecular
formula $C_8H_7NO_2$ has been observed in the mass spectrum in an experimental study (Montoya-Aguilera et al., 2017),
supporting the formation of the predicted N-(2-formylphenyl)formamide. To the best of our knowledge, this study is the first
to reveal the chemical identity of the mass spectrum signal as N-(2-formylphenyl)formamide, as opposed to the proposed 3-
oxy-2-hydroxy-indole. In addition, monomeric products (isatin and isatoic anhydride) and dimer products has not been
observed in our computational study. We speculate that they may be produced from the subsequent conversion of the formed
alkoxy radicals, multi-generation reactions of organonitrates and hydroperoxide and cross reactions of peroxy radicals (RO$_2$ +
RO$_2$). Therefore, further studies are warranted to investigate the subsequent transformation of the formed alkoxy radicals,
organonitrates and hydroperoxide, and the RO$_2$ + RO$_2$ reactions, to accurately describe the atmospheric impact of indole.
The calculated $k_{Cl}$ value of the indole + ·Cl reaction is a factor of 3.7 higher than that of the indole + ·OH reaction, and is
close to the $k_{Cl}$ values for the reactions of alkylamines, heterocyclic amines and amides with ·Cl (Xie et al., 2017; Xie et al.,
2015; Ma et al., 2018a; Nicovich et al., 2015). The contribution of ·Cl to the transformation of indole is calculated to be 3.6-
36% that of ·OH, assuming ·Cl concentrations equal to 1-10% of that of ·OH (Wang and Ruiz, 2017; Nicovich et al., 2015;
Xie et al., 2017; Xie et al., 2015; Ma et al., 2018a). Therefore, ·Cl plays an important role in the overall transformation of
indole. More importantly, ·Cl initiated reaction of indole does not lead to the formation of carcinogenic nitrosamines although
·Cl can favorably abstract the H-atom from N-site to form $C_8H_6N$ radicals, which is a plausible precursor of carcinogenic
nitrosamines. Hence, to the best of our knowledge, this is the first study to reveal despite forming radicals by abstracting an
H-atom at the N-site, carcinogenic nitrosamines were not produced in the indole oxidation reaction. This is most likely caused
by the delocalized character of the formed $C_8H_6N$ radicals due to the existence of the adjacent unsaturated bonds. Therefore,
this study further confirm that the functional groups connected to the NH$_x$ ($x$ = 1, 2) group highly affect the atmospheric fate
of ONCs. Further studies should be performed to investigate the structure-activity relationship of ·Cl initiated reactions of
ONCs to comprehensively evaluate their atmospheric impacts.

*Data availability.* The data in this article are available from the corresponding author upon request (maff@dlut.edu.cn,
hbxie@dlut.edu.cn).
*Author contribution.* FFM and HBX designed research; JWX, FFM and HBX performed research; JWX, FFM and HBX
analyzed data; JWX, FFM, HBX and JWC wrote the paper; and FFM, HBX and JWC reviewed and revised the paper.
*Competing interests.* The authors declare that they have no conflict of interest.



*Acknowledgements.* The study was supported by the LiaoNing Revitalization Talents Program (XLYC1907194), National
Natural Science Foundation of China (21876024), the Major International (Regional) Joint Research Project (21661142001)
and Supercomputing Center of Dalian University of Technology.

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
