# Peer review of "Atmospheric Oxidation Mechanism and Kinetics of Indole Initiated by ·OH and ·Cl: A Computational Study"

_Atmospheric Chemistry and Physics, 2022_

## Author Comment (AC1)

Table 1. Calculated reaction rate constants ($k$) at 298 K and over the pressure range from 0.1 to 1.0 atm of the main reaction pathways for the indole + ·OH/·Cl reactions

| Pathways | $k$ | | | |
|---|---|---|---|---|
| | 0.1 atm | 0.4 atm | 0.7 atm | 1.0 atm |
| Indole + ·OH | $7.90 \times 10^{-11}$ cm$^3$ molecule$^{-1}$ s$^{-1}$ | $7.90 \times 10^{-11}$ cm$^3$ molecule$^{-1}$ s$^{-1}$ | $7.90 \times 10^{-11}$ cm$^3$ molecule$^{-1}$ s$^{-1}$ | $7.90 \times 10^{-11}$ cm$^3$ molecule$^{-1}$ s$^{-1}$ |
| Indole + ·Cl | $2.91 \times 10^{-10}$ cm$^3$ molecule$^{-1}$ s$^{-1}$ | $2.91 \times 10^{-10}$ cm$^3$ molecule$^{-1}$ s$^{-1}$ | $2.91 \times 10^{-10}$ cm$^3$ molecule$^{-1}$ s$^{-1}$ | $2.91 \times 10^{-10}$ cm$^3$ molecule$^{-1}$ s$^{-1}$ |
| IM$_{1-7}$ + O$_2$ | $6.12 \times 10^{-12}$ cm$^3$ molecule$^{-1}$ s$^{-1}$ | $6.12 \times 10^{-12}$ cm$^3$ molecule$^{-1}$ s$^{-1}$ | $6.12 \times 10^{-12}$ cm$^3$ molecule$^{-1}$ s$^{-1}$ | $6.12 \times 10^{-12}$ cm$^3$ molecule$^{-1}$ s$^{-1}$ |
| IM$_{2-5}$ + O$_2$ | $6.15 \times 10^{-12}$ cm$^3$ molecule$^{-1}$ s$^{-1}$ | $6.15 \times 10^{-12}$ cm$^3$ molecule$^{-1}$ s$^{-1}$ | $6.15 \times 10^{-12}$ cm$^3$ molecule$^{-1}$ s$^{-1}$ | $6.15 \times 10^{-12}$ cm$^3$ molecule$^{-1}$ s$^{-1}$ |
| IM$_{2-6}$ + O$_2$ | $6.10 \times 10^{-12}$ cm$^3$ molecule$^{-1}$ s$^{-1}$ | $6.10 \times 10^{-12}$ cm$^3$ molecule$^{-1}$ s$^{-1}$ | $6.10 \times 10^{-12}$ cm$^3$ molecule$^{-1}$ s$^{-1}$ | $6.10 \times 10^{-12}$ cm$^3$ molecule$^{-1}$ s$^{-1}$ |
| C$_8$H$_6$N + O$_2$ | $6.13 \times 10^{-12}$ cm$^3$ molecule$^{-1}$ s$^{-1}$ | $6.13 \times 10^{-12}$ cm$^3$ molecule$^{-1}$ s$^{-1}$ | $6.13 \times 10^{-12}$ cm$^3$ molecule$^{-1}$ s$^{-1}$ | $6.13 \times 10^{-12}$ cm$^3$ molecule$^{-1}$ s$^{-1}$ |
| IM$_{1-7}$-4OO-$s\rightarrow$ IM$_{1-7}$-4OO-OH-$s$ | $1.22 \times 10^{-2}$ s$^{-1}$ | $1.22 \times 10^{-2}$ s$^{-1}$ | $1.22 \times 10^{-2}$ s$^{-1}$ | $1.22 \times 10^{-2}$ s$^{-1}$ |
| IM$_{2-5}$-6OO-$a\rightarrow$ IM$_{2-5}$-6OO-C5H-$a$ | $7.65 \times 10^{-4}$ s$^{-1}$ | $7.65 \times 10^{-4}$ s$^{-1}$ | $7.65 \times 10^{-4}$ s$^{-1}$ | $7.65 \times 10^{-4}$ s$^{-1}$ |
| IM$_{2-6}$-5OO-$a\rightarrow$ IM$_{2-6}$-52OO-$a$ | $3.60 \times 10^{-7}$ s$^{-1}$ | $3.60 \times 10^{-7}$ s$^{-1}$ | $3.60 \times 10^{-7}$ s$^{-1}$ | $3.60 \times 10^{-7}$ s$^{-1}$ |
| C$_8$H$_6$N-4OO-$a/s\rightarrow$ C$_8$H$_6$N-43OO-$a/s$ | $8.77 \times 10^{-9}$ s$^{-1}$ | $8.77 \times 10^{-9}$ s$^{-1}$ | $8.77 \times 10^{-9}$ s$^{-1}$ | $8.77 \times 10^{-9}$ s$^{-1}$ |

Table 2. Calculated reaction rate constants ($k$) at 298 K and over the energy transfer parameters from 50 to 250 cm$^{-1}$ of the main reaction pathways for the indole + ·OH/·Cl reactions

| Pathways | $k$ | | | | |
|---|---|---|---|---|---|
| | $\Delta E_d = 50$ cm$^{-1}$ | $\Delta E_d = 100$ cm$^{-1}$ | $\Delta E_d = 150$ cm$^{-1}$ | $\Delta E_d = 200$ cm$^{-1}$ | $\Delta E_d = 250$ cm$^{-1}$ |
| Indole+·OH | $7.89 \times 10^{-11}$ cm$^3$ molecule$^{-1}$ s$^{-1}$ | $7.89 \times 10^{-11}$ cm$^3$ molecule$^{-1}$ s$^{-1}$ | $7.90 \times 10^{-11}$ cm$^3$ molecule$^{-1}$ s$^{-1}$ | $7.90 \times 10^{-11}$ cm$^3$ molecule$^{-1}$ s$^{-1}$ | $7.90 \times 10^{-11}$ cm$^3$ molecule$^{-1}$ s$^{-1}$ |
| Indole+·Cl | $2.90 \times 10^{-10}$ cm$^3$ molecule$^{-1}$ s$^{-1}$ | $2.90 \times 10^{-10}$ cm$^3$ molecule$^{-1}$ s$^{-1}$ | $2.91 \times 10^{-10}$ cm$^3$ molecule$^{-1}$ s$^{-1}$ | $2.91 \times 10^{-10}$ cm$^3$ molecule$^{-1}$ s$^{-1}$ | $2.91 \times 10^{-10}$ cm$^3$ molecule$^{-1}$ s$^{-1}$ |
| IM$_{1-7}$ + O$_2$ | $6.12 \times 10^{-12}$ cm$^3$ molecule$^{-1}$ s$^{-1}$ | $6.12 \times 10^{-12}$ cm$^3$ molecule$^{-1}$ s$^{-1}$ | $6.12 \times 10^{-12}$ cm$^3$ molecule$^{-1}$ s$^{-1}$ | $6.12 \times 10^{-12}$ cm$^3$ molecule$^{-1}$ s$^{-1}$ | $6.12 \times 10^{-12}$ cm$^3$ molecule$^{-1}$ s$^{-1}$ |
| IM$_{2-5}$ + O$_2$ | $6.15 \times 10^{-12}$ cm$^3$ molecule$^{-1}$ s$^{-1}$ | $6.15 \times 10^{-12}$ cm$^3$ molecule$^{-1}$ s$^{-1}$ | $6.15 \times 10^{-12}$ cm$^3$ molecule$^{-1}$ s$^{-1}$ | $6.15 \times 10^{-12}$ cm$^3$ molecule$^{-1}$ s$^{-1}$ | $6.15 \times 10^{-12}$ cm$^3$ molecule$^{-1}$ s$^{-1}$ |
| IM$_{2-6}$ + O$_2$ | $6.10 \times 10^{-12}$ cm$^3$ molecule$^{-1}$ s$^{-1}$ | $6.10 \times 10^{-12}$ cm$^3$ molecule$^{-1}$ s$^{-1}$ | $6.10 \times 10^{-12}$ cm$^3$ molecule$^{-1}$ s$^{-1}$ | $6.10 \times 10^{-12}$ cm$^3$ molecule$^{-1}$ s$^{-1}$ | $6.10 \times 10^{-12}$ cm$^3$ molecule$^{-1}$ s$^{-1}$ |
| C$_8$H$_6$N + O$_2$ | $6.13 \times 10^{-12}$ cm$^3$ molecule$^{-1}$ s$^{-1}$ | $6.13 \times 10^{-12}$ cm$^3$ molecule$^{-1}$ s$^{-1}$ | $6.13 \times 10^{-12}$ cm$^3$ molecule$^{-1}$ s$^{-1}$ | $6.13 \times 10^{-12}$ cm$^3$ molecule$^{-1}$ s$^{-1}$ | $6.13 \times 10^{-12}$ cm$^3$ molecule$^{-1}$ s$^{-1}$ |
| IM$_{1-7}$-4OO-$s\rightarrow$IM$_{1-7}$-4OO-OH-$s$ | $1.22 \times 10^{-2}$ s$^{-1}$ | $1.22 \times 10^{-2}$ s$^{-1}$ | $1.22 \times 10^{-2}$ s$^{-1}$ | $1.22 \times 10^{-2}$ s$^{-1}$ | $1.22 \times 10^{-2}$ s$^{-1}$ |
| IM$_{2-5}$-6OO-$a\rightarrow$IM$_{2-5}$-6OO-C5H-$a$ | $7.65 \times 10^{-4}$ s$^{-1}$ | $7.65 \times 10^{-4}$ s$^{-1}$ | $7.65 \times 10^{-4}$ s$^{-1}$ | $7.65 \times 10^{-4}$ s$^{-1}$ | $7.65 \times 10^{-4}$ s$^{-1}$ |
| IM$_{2-6}$-5OO-$a\rightarrow$IM$_{2-6}$-52OO-$a$ | $3.60 \times 10^{-7}$ s$^{-1}$ | $3.60 \times 10^{-7}$ s$^{-1}$ | $3.60 \times 10^{-7}$ s$^{-1}$ | $3.60 \times 10^{-7}$ s$^{-1}$ | $3.60 \times 10^{-7}$ s$^{-1}$ |
| C$_8$H$_6$N-4OO-$a/s\rightarrow$ C$_8$H$_6$N-43OO-$a/s$ | $8.77 \times 10^{-9}$ s$^{-1}$ | $8.77 \times 10^{-9}$ s$^{-1}$ | $8.77 \times 10^{-9}$ s$^{-1}$ | $8.77 \times 10^{-9}$ s$^{-1}$ | $8.77 \times 10^{-9}$ s$^{-1}$ |

Table 3. Calculated branching ratios ($\Gamma$) at 298 K and over the pressure range from 0.1 to 1.0 atm of the main reaction pathways for the indole + $\cdot$OH/$\cdot$Cl reactions

| Species | $\Gamma$ | | | |
|---|---|---|---|---|
| | 0.1 atm | 0.4 atm | 0.7 atm | 1.0 atm |
| IM$_{1-7}$ | 77.4% | 77.4% | 77.4% | 77.4% |
| IM$_{2-5}$ | 31.4% | 31.4% | 31.4% | 31.4% |
| IM$_{2-6}$ | 45.5% | 45.5% | 45.5% | 45.5% |
| P$_{2-10}$ | 23.1% | 23.1% | 23.1% | 23.1% |
| P$_{1-7-4-1}$ | 6.6% | 6.5% | 6.5% | 6.5% |
| NO-P$_3$ | 67.3% | 67.3% | 67.3% | 67.3% |
| HO$_2$-P$_3$ | 24.9% | 24.9% | 24.9% | 24.9% |
| NO-P$_4$ | 72.4% | 72.4% | 72.4% | 72.4% |
| HO$_2$-P$_4$ | 26.8% | 26.8% | 26.8% | 26.8% |
| NO-P$_5$ | 72.7% | 72.7% | 72.7% | 72.7% |
| HO$_2$-P$_5$ | 26.9% | 26.9% | 26.9% | 26.9% |
| NO-P$_6$ | 73.0% | 73.0% | 73.0% | 73.0% |
| HO$_2$-P$_6$ | 27.0% | 27.0% | 27.0% | 27.0% |

Table 4. Calculated branching ratios ($\Gamma$) at 298 K and over the energy transfer parameters from 50 to 250 cm$^{-1}$ of the main reaction pathways for the indole + $\cdot$OH/$\cdot$Cl reactions

| Species | $\Gamma$ | | | | |
|---|---|---|---|---|---|
| | $\Delta E_d = 50$ cm$^{-1}$ | $\Delta E_d = 100$ cm$^{-1}$ | $\Delta E_d = 150$ cm$^{-1}$ | $\Delta E_d = 200$ cm$^{-1}$ | $\Delta E_d = 250$ cm$^{-1}$ |
| IM$_{1-7}$ | 77.4% | 77.4% | 77.4% | 77.4% | 77.4% |
| IM$_{2-5}$ | 31.4% | 31.4% | 31.4% | 31.4% | 31.4% |
| IM$_{2-6}$ | 45.5% | 45.5% | 45.5% | 45.5% | 45.5% |
| P$_{2-10}$ | 23.1% | 23.1% | 23.1% | 23.1% | 23.1% |
| P$_{1-7-4-1}$ | 6.5% | 6.5% | 6.5% | 6.5% | 6.5% |
| NO-P$_3$ | 67.3% | 67.3% | 67.3% | 67.3% | 67.3% |
| HO$_2$-P$_3$ | 24.9% | 24.9% | 24.9% | 24.9% | 24.9% |
| NO-P$_4$ | 72.4% | 72.4% | 72.4% | 72.4% | 72.4% |
| HO$_2$-P$_4$ | 26.8% | 26.8% | 26.8% | 26.8% | 26.8% |
| NO-P$_5$ | 72.7% | 72.7% | 72.7% | 72.7% | 72.7% |
| HO$_2$-P$_5$ | 26.9% | 26.9% | 26.9% | 26.9% | 26.9% |
| NO-P$_6$ | 73.0% | 73.0% | 73.0% | 73.0% | 73.0% |
| HO$_2$-P$_6$ | 27.0% | 27.0% | 27.0% | 27.0% | 27.0% |

---

## Author Response (AR1)

The authors would like to thank the reviewers for this discussion and their constructive comments and suggestions. We have carefully replied to all their comments and have made improvements to the paper based on their suggestions.

Reviewer 1

**General comment**

This manuscript presents theoretical calculations on the mechanisms and kinetics of the reaction systems initiated by the indole + OH and the indole + Cl reaction under atmospheric conditions. In particular, consecutive reactions of the produced intermediate radicals with $O_2$ were included, and the unimolecular reactions of the subsequent radical-$O_2$ adducts were studied in their competition to bimolecular reactions with NO and $HO_2$. Relative yields for the most important reaction channels were derived. An important finding is that the N-centered radicals produced by hydrogen abstraction from the indole nitrogen react much faster with $O_2$ than with NO under typical tropospheric conditions. It is concluded that the formation of carcinogenic nitrosamines appears less important for indole as for aliphatic amines (at least via this reaction pathway). The detailed theoretical characterization of several very complex reaction mechanisms with advanced quantum chemical calculations and statistical rate theory must have been really painstaking work. As the problem addressed in this paper is timely, and the methods applied appear adequate, the manuscript merits publication. However, before final acceptance, the authors could further improve the quality by considering a few minor points.

**Response:** We appreciate the positive comments and have revised the manuscript to further enhance its quality.

**Special Suggestions and Comments**

(1) They should make a bit more clear (in section 2.2) why two different software packages (MultiWell and MESMER) were used. Which one was used for which reaction, and why?

**Response:** We appreciated the suggestion and agree that the statement is not clear here. In the revised manuscript, the sentence has changed to:

"MultiWell-2014.1 and MESMER 5.0 software were employed to investigate the kinetics for short time and long time reaction, respectively (Barker and Ortiz, 2001; Barker, 2001; Glowacki et al., 2012). For the initial reactions of ·OH/·Cl + indole, the reaction kinetics were calculated within the MultiWell-2014.1 program. For the subsequent reactions of resulting primary intermediates, MESMER 5.0 were selected for simulating the reaction kinetics, since it has good performance for long time runs, especially for simulating the variation of the different intermediates over time." (Lines 106-110.)

(2) Please explicitly state whether you included all the channels shown in Fig. 1(A) for OH and in Fig. 1(B) for Cl in the respective master equations. In other words, did you use a full multichannel approach coupling the entire reaction system or did you solve a

corresponding number of one(few)-channel master equations?

**Response:** Thanks for the suggestion. The suggestion has been adopted. The original sentence was changed to:

"With the master equation theory, the overall rate constants ($k_{OH}$ and $k_{Cl}$) and branching ratios ($\Gamma$) for all H-abstraction and ·OH/·Cl-addition pathways involved in the ·OH/·Cl + indole reactions were calculated at 298 K and 1 atm." (Lines 166-167.)

(3) Furthermore, as far as this reviewer understands, the calculations were obviously performed for a single pressure of 1 atm. Does pressure have any effect on the relative yields in the tropospherically relevant range down to say 100 mbar? And if so, what about the energy transfer parameters (e.g. the average energy transferred per collision)? After all, how do the time-dependent results from the master equation calculations (like those illustrated in Fig. 3) translate to steady-state situations in the atmosphere. Here a brief discussion would also be useful.

**Response:** We appreciated the suggestion. We have tested the effect of pressure and average transfer energy on the reaction rate constants and branching ratios of the main reaction pathways for the ·OH/·Cl + indole reactions. The pressure in the range of 0.1 to 1 atm and the average energy transfer in the range of 50 to 250 cm$^{-1}$ were selected to test the effect. In the revised manuscript, the following sentences were added:

"In addition, $\Delta E_d$ between 50 - 250 cm$^{-1}$ were selected to study energy transfer parameters effects.

The kinetic calculations were primarily performed at 298 K and 1 atm, with additional ones at 0.1, 0.4 and 0.7 atm in the troposphere relevant range to explore pressure effects. Variation of the energy transfer parameters and pressure resulted in only minor changes (< 0.1%) in the calculated rate coefficients and branching ratios of main reaction pathways (see details in the SI)." (Lines 120 and 123-126.)

In addition, we thought a large macroscopic model would be required to assess the impact on steady state species under atmospheric conditions with a continuous source of indole. We have to acknowledge that such simulation could be more accurate than our current simulation. However, this is beyond the scope of the present work. Our work can yield rate coefficients that can be used in future macroscopic simulations.

Table S6. Calculated reaction rate constants ($k$) at 298 K and over the pressure range from 0.1 to 1.0 atm of the main reaction pathways for the indole + ·OH/·Cl reactions

| Pathways | $k$ | | | |
|---|---|---|---|---|
| | 0.1 atm | 0.4 atm | 0.7 atm | 1.0 atm |
| Indole + ·OH | $7.90 \times 10^{-11}$ cm$^3$ molecule$^{-1}$ s$^{-1}$ | $7.90 \times 10^{-11}$ cm$^3$ molecule$^{-1}$ s$^{-1}$ | $7.90 \times 10^{-11}$ cm$^3$ molecule$^{-1}$ s$^{-1}$ | $7.90 \times 10^{-11}$ cm$^3$ molecule$^{-1}$ s$^{-1}$ |
| Indole + ·Cl | $2.91 \times 10^{-10}$ cm$^3$ molecule$^{-1}$ s$^{-1}$ | $2.91 \times 10^{-10}$ cm$^3$ molecule$^{-1}$ s$^{-1}$ | $2.91 \times 10^{-10}$ cm$^3$ molecule$^{-1}$ s$^{-1}$ | $2.91 \times 10^{-10}$ cm$^3$ molecule$^{-1}$ s$^{-1}$ |
| IM$_{1-7}$ + O$_2$ | $6.12 \times 10^{-12}$ cm$^3$ molecule$^{-1}$ s$^{-1}$ | $6.12 \times 10^{-12}$ cm$^3$ molecule$^{-1}$ s$^{-1}$ | $6.12 \times 10^{-12}$ cm$^3$ molecule$^{-1}$ s$^{-1}$ | $6.12 \times 10^{-12}$ cm$^3$ molecule$^{-1}$ s$^{-1}$ |
| IM$_{2-5}$ + O$_2$ | $6.15 \times 10^{-12}$ cm$^3$ molecule$^{-1}$ s$^{-1}$ | $6.15 \times 10^{-12}$ cm$^3$ molecule$^{-1}$ s$^{-1}$ | $6.15 \times 10^{-12}$ cm$^3$ molecule$^{-1}$ s$^{-1}$ | $6.15 \times 10^{-12}$ cm$^3$ molecule$^{-1}$ s$^{-1}$ |
| IM$_{2-6}$ + O$_2$ | $6.10 \times 10^{-12}$ cm$^3$ molecule$^{-1}$ s$^{-1}$ | $6.10 \times 10^{-12}$ cm$^3$ molecule$^{-1}$ s$^{-1}$ | $6.10 \times 10^{-12}$ cm$^3$ molecule$^{-1}$ s$^{-1}$ | $6.10 \times 10^{-12}$ cm$^3$ molecule$^{-1}$ s$^{-1}$ |
| C$_8$H$_6$N + O$_2$ | $6.13 \times 10^{-12}$ cm$^3$ molecule$^{-1}$ s$^{-1}$ | $6.13 \times 10^{-12}$ cm$^3$ molecule$^{-1}$ s$^{-1}$ | $6.13 \times 10^{-12}$ cm$^3$ molecule$^{-1}$ s$^{-1}$ | $6.13 \times 10^{-12}$ cm$^3$ molecule$^{-1}$ s$^{-1}$ |
| IM$_{1-7}$-4OO-$s$→ IM$_{1-7}$-4OO-OH-$s$ | $1.22 \times 10^{-2}$ s$^{-1}$ | $1.22 \times 10^{-2}$ s$^{-1}$ | $1.22 \times 10^{-2}$ s$^{-1}$ | $1.22 \times 10^{-2}$ s$^{-1}$ |
| IM$_{2-5}$-6OO-$a$→ IM$_{2-5}$-6OO-C5H-$a$ | $7.65 \times 10^{-4}$ s$^{-1}$ | $7.65 \times 10^{-4}$ s$^{-1}$ | $7.65 \times 10^{-4}$ s$^{-1}$ | $7.65 \times 10^{-4}$ s$^{-1}$ |
| IM$_{2-6}$-5OO-$a$→ IM$_{2-6}$-52OO-$a$ | $3.60 \times 10^{-7}$ s$^{-1}$ | $3.60 \times 10^{-7}$ s$^{-1}$ | $3.60 \times 10^{-7}$ s$^{-1}$ | $3.60 \times 10^{-7}$ s$^{-1}$ |
| C$_8$H$_6$N-4OO-$a/s$→ C$_8$H$_6$N-43OO-$a/s$ | $8.77 \times 10^{-9}$ s$^{-1}$ | $8.77 \times 10^{-9}$ s$^{-1}$ | $8.77 \times 10^{-9}$ s$^{-1}$ | $8.77 \times 10^{-9}$ s$^{-1}$ |

Table S7. Calculated reaction rate constants ($k$) at 298 K and over the energy transfer parameters from 50 to 250 cm$^{-1}$ of the main reaction pathways for the indole + ·OH/·Cl reactions

| Pathways | $k$ | | | | |
|---|---|---|---|---|---|
| | $\Delta E_d = 50$ cm$^{-1}$ | $\Delta E_d = 100$ cm$^{-1}$ | $\Delta E_d = 150$ cm$^{-1}$ | $\Delta E_d = 200$ cm$^{-1}$ | $\Delta E_d = 250$ cm$^{-1}$ |
| Indole+·OH | $7.89 \times 10^{-11}$ cm$^3$ molecule$^{-1}$ s$^{-1}$ | $7.89 \times 10^{-11}$ cm$^3$ molecule$^{-1}$ s$^{-1}$ | $7.90 \times 10^{-11}$ cm$^3$ molecule$^{-1}$ s$^{-1}$ | $7.90 \times 10^{-11}$ cm$^3$ molecule$^{-1}$ s$^{-1}$ | $7.90 \times 10^{-11}$ cm$^3$ molecule$^{-1}$ s$^{-1}$ |
| Indole+·Cl | $2.90 \times 10^{-10}$ cm$^3$ molecule$^{-1}$ s$^{-1}$ | $2.90 \times 10^{-10}$ cm$^3$ molecule$^{-1}$ s$^{-1}$ | $2.91 \times 10^{-10}$ cm$^3$ molecule$^{-1}$ s$^{-1}$ | $2.91 \times 10^{-10}$ cm$^3$ molecule$^{-1}$ s$^{-1}$ | $2.91 \times 10^{-10}$ cm$^3$ molecule$^{-1}$ s$^{-1}$ |
| IM$_{1-7}$ + O$_2$ | $6.12 \times 10^{-12}$ cm$^3$ molecule$^{-1}$ s$^{-1}$ | $6.12 \times 10^{-12}$ cm$^3$ molecule$^{-1}$ s$^{-1}$ | $6.12 \times 10^{-12}$ cm$^3$ molecule$^{-1}$ s$^{-1}$ | $6.12 \times 10^{-12}$ cm$^3$ molecule$^{-1}$ s$^{-1}$ | $6.12 \times 10^{-12}$ cm$^3$ molecule$^{-1}$ s$^{-1}$ |
| IM$_{2-5}$ + O$_2$ | $6.15 \times 10^{-12}$ cm$^3$ molecule$^{-1}$ s$^{-1}$ | $6.15 \times 10^{-12}$ cm$^3$ molecule$^{-1}$ s$^{-1}$ | $6.15 \times 10^{-12}$ cm$^3$ molecule$^{-1}$ s$^{-1}$ | $6.15 \times 10^{-12}$ cm$^3$ molecule$^{-1}$ s$^{-1}$ | $6.15 \times 10^{-12}$ cm$^3$ molecule$^{-1}$ s$^{-1}$ |
| IM$_{2-6}$ + O$_2$ | $6.10 \times 10^{-12}$ cm$^3$ molecule$^{-1}$ s$^{-1}$ | $6.10 \times 10^{-12}$ cm$^3$ molecule$^{-1}$ s$^{-1}$ | $6.10 \times 10^{-12}$ cm$^3$ molecule$^{-1}$ s$^{-1}$ | $6.10 \times 10^{-12}$ cm$^3$ molecule$^{-1}$ s$^{-1}$ | $6.10 \times 10^{-12}$ cm$^3$ molecule$^{-1}$ s$^{-1}$ |
| C$_8$H$_6$N + O$_2$ | $6.13 \times 10^{-12}$ cm$^3$ molecule$^{-1}$ s$^{-1}$ | $6.13 \times 10^{-12}$ cm$^3$ molecule$^{-1}$ s$^{-1}$ | $6.13 \times 10^{-12}$ cm$^3$ molecule$^{-1}$ s$^{-1}$ | $6.13 \times 10^{-12}$ cm$^3$ molecule$^{-1}$ s$^{-1}$ | $6.13 \times 10^{-12}$ cm$^3$ molecule$^{-1}$ s$^{-1}$ |
| IM$_{1-7}$-4OO-$s$→ IM$_{1-7}$-4OO-OH-$s$ | $1.22 \times 10^{-2}$ s$^{-1}$ | $1.22 \times 10^{-2}$ s$^{-1}$ | $1.22 \times 10^{-2}$ s$^{-1}$ | $1.22 \times 10^{-2}$ s$^{-1}$ | $1.22 \times 10^{-2}$ s$^{-1}$ |
| IM$_{2-5}$-6OO-$a$→ IM$_{2-5}$-6OO-C5H-$a$ | $7.65 \times 10^{-4}$ s$^{-1}$ | $7.65 \times 10^{-4}$ s$^{-1}$ | $7.65 \times 10^{-4}$ s$^{-1}$ | $7.65 \times 10^{-4}$ s$^{-1}$ | $7.65 \times 10^{-4}$ s$^{-1}$ |
| IM$_{2-6}$-5OO-$a$→ IM$_{2-6}$-52OO-$a$ | $3.60 \times 10^{-7}$ s$^{-1}$ | $3.60 \times 10^{-7}$ s$^{-1}$ | $3.60 \times 10^{-7}$ s$^{-1}$ | $3.60 \times 10^{-7}$ s$^{-1}$ | $3.60 \times 10^{-7}$ s$^{-1}$ |
| C$_8$H$_6$N-4OO-$a/s$→ C$_8$H$_6$N-43OO-$a/s$ | $8.77 \times 10^{-9}$ s$^{-1}$ | $8.77 \times 10^{-9}$ s$^{-1}$ | $8.77 \times 10^{-9}$ s$^{-1}$ | $8.77 \times 10^{-9}$ s$^{-1}$ | $8.77 \times 10^{-9}$ s$^{-1}$ |

Table S8. Calculated branching ratios ($\Gamma$) at 298 K and over the pressure range from 0.1 to 1.0 atm of the main reaction pathways for the indole + ·OH/·Cl reactions

| Species | $\Gamma$ | | | |
|---|---|---|---|---|
| | 0.1 atm | 0.4 atm | 0.7 atm | 1.0 atm |
| IM$_{1-7}$ | 77.4% | 77.4% | 77.4% | 77.4% |
| IM$_{2-5}$ | 31.4% | 31.4% | 31.4% | 31.4% |
| IM$_{2-6}$ | 45.5% | 45.5% | 45.5% | 45.5% |
| P$_{2-10}$ | 23.1% | 23.1% | 23.1% | 23.1% |
| P$_{1-7-4-1}$ | 6.6% | 6.5% | 6.5% | 6.5% |
| NO-P$_3$ | 67.3% | 67.3% | 67.3% | 67.3% |
| HO$_2$-P$_3$ | 24.9% | 24.9% | 24.9% | 24.9% |
| NO-P$_4$ | 72.4% | 72.4% | 72.4% | 72.4% |
| HO$_2$-P$_4$ | 26.8% | 26.8% | 26.8% | 26.8% |
| NO-P$_5$ | 72.7% | 72.7% | 72.7% | 72.7% |
| HO$_2$-P$_5$ | 26.9% | 26.9% | 26.9% | 26.9% |
| NO-P$_6$ | 73.0% | 73.0% | 73.0% | 73.0% |
| HO$_2$-P$_6$ | 27.0% | 27.0% | 27.0% | 27.0% |

Table S9. Calculated branching ratios ($\Gamma$) at 298 K and over the energy transfer parameters from 50 to 250 cm$^{-1}$ of the main reaction pathways for the indole + ·OH/·Cl reactions

| Species | $\Gamma$ | | | | |
|---|---|---|---|---|---|
| | $\Delta E_d = 50$ cm$^{-1}$ | $\Delta E_d = 100$ cm$^{-1}$ | $\Delta E_d = 150$ cm$^{-1}$ | $\Delta E_d = 200$ cm$^{-1}$ | $\Delta E_d = 250$ cm$^{-1}$ |
| IM$_{1-7}$ | 77.4% | 77.4% | 77.4% | 77.4% | 77.4% |
| IM$_{2-5}$ | 31.4% | 31.4% | 31.4% | 31.4% | 31.4% |
| IM$_{2-6}$ | 45.5% | 45.5% | 45.5% | 45.5% | 45.5% |
| P$_{2-10}$ | 23.1% | 23.1% | 23.1% | 23.1% | 23.1% |
| P$_{1-7-4-1}$ | 6.5% | 6.5% | 6.5% | 6.5% | 6.5% |
| NO-P$_3$ | 67.3% | 67.3% | 67.3% | 67.3% | 67.3% |
| HO$_2$-P$_3$ | 24.9% | 24.9% | 24.9% | 24.9% | 24.9% |
| NO-P$_4$ | 72.4% | 72.4% | 72.4% | 72.4% | 72.4% |
| HO$_2$-P$_4$ | 26.8% | 26.8% | 26.8% | 26.8% | 26.8% |
| NO-P$_5$ | 72.7% | 72.7% | 72.7% | 72.7% | 72.7% |
| HO$_2$-P$_5$ | 26.9% | 26.9% | 26.9% | 26.9% | 26.9% |
| NO-P$_6$ | 73.0% | 73.0% | 73.0% | 73.0% | 73.0% |
| HO$_2$-P$_6$ | 27.0% | 27.0% | 27.0% | 27.0% | 27.0% |

(4) line 9: please insert comma after nitrosamines.
**Response:** We have fixed this. (Line 9.)

(5) line 14: please insert 'the' before dominant.
**Response:** We have fixed this. (Line 14.)

(6) line 21: please insert 'that' after demonstrate
**Response:** We have fixed this. (Line 21.)

(7) line 34: '10% of total gas phase nitrogen' – probably excluding $N_2$.
**Response:** Yes. We changed the sentence to:
"Until now, about 160 ONCs have been detected in the atmosphere, accounting for 10% of total gas phase nitrogen (excluding $N_2$)" (Lines 34-35.)

(8) line 60: 'the $k_{OH}$ value' should be correctly termed rate constant.
**Response:** We have fixed this. (Line 60.)

(9) line 78: 'reactions' should better read 'reaction'.
**Response:** We have fixed this. (Line 78.)

(10) line 123: 'phenyl group' should probably better read 'the benzene ring' or 'the C6 ring'.
**Response:** We have used "benzene ring" instead of "phenyl group'. (Line 129.)

(11) line 126: The authors should mention at this point that the numbering of the atoms is given in Fig. 1.
**Response:** We have revised the sentence. (Lines 132-133.)

(12) line 239: 'biomolecular' read 'bimolecular'.
**Response:** We have corrected it. (Line 246.)

(13) line 283: please insert 'that' after reveal (also line 250).
**Response:** According to the reviewer's suggestions, the corresponding corrections were done. (Lines 257 and 291.)

(14) line 292: If you specify author contributions then please also include Jonas Elm.
**Response:** We agree that the author contribution from Jonas Elm should be included. We have added it. (Line 301.)

(15) Fig. S1: Please mention the pressure in the figure caption
**Response:** We have added the pressure in the figure caption. (Figure S1.)

**Reviewer 2**

**Reviewer: General comment**

I have no further comments.

**Response:** Thanks for the positive comment. We have revised the manuscript to further enhance its quality.